# Investigating the characteristics and correlates of systemic inflammation after traumatic brain injury: the TBI-BraINFLAMM study

Lucia M Li ,[1,2] Amanda Heslegrave,[3,4] Eyal Soreq,[1,2] Giovanni Nattino,[5] Margherita Rosnati,[1,6] Elena Garbero ,[7] Karl A Zimmerman,[1,8] Neil S N Graham ,[1,2] Federico Moro,[9] Deborah Novelli,[10] Primoz Gradisek,[11,12] Sandra Magnoni,[13] Ben Glocker,[6] Henrik Zetterberg,[4,14,15] Guido Bertolini,[16] David J Sharp[2,17]

For numbered affiliations see end of article.

Correspondence to
Dr Lucia M Li;
lml34@doctors.org.uk

## ABSTRACT

**Introduction** A significant environmental risk factor for neurodegenerative disease is traumatic brain injury (TBI). However, it is not clear how TBI results in ongoing chronic neurodegeneration. Animal studies show that systemic inflammation is signalled to the brain. This can result in sustained and aggressive microglial activation, which in turn is associated with widespread neurodegeneration. We aim to evaluate systemic inflammation as a mediator of ongoing neurodegeneration after TBI.

**Methods and analysis** TBI-braINFLAMM will combine data already collected from two large prospective TBI studies. The CREACTIVE study, a broad consortium which enrolled >8000 patients with TBI to have CT scans and blood samples in the hyperacute period, has data available from 854 patients. The BIO-AX-TBI study recruited 311 patients to have acute CT scans, longitudinal blood samples and longitudinal MRI brain scans. The BIO-AX-TBI study also has data from 102 healthy and 24 non-TBI trauma controls, comprising blood samples (both control groups) and MRI scans (healthy controls only). All blood samples from BIO-AX-TBI and CREACTIVE have already been tested for neuronal injury markers (GFAP, tau and NfL), and CREACTIVE blood samples have been tested for inflammatory cytokines. We will additionally test inflammatory cytokine levels from the already collected longitudinal blood samples in the BIO-AX-TBI study, as well as matched microdialysate and blood samples taken during the acute period from a subgroup of patients with TBI (n=18).

We will use this unique dataset to characterise post-TBI systemic inflammation, and its relationships with injury severity and ongoing neurodegeneration.

**Ethics and dissemination** Ethical approval for this study has been granted by the London—Camberwell St Giles Research Ethics Committee (17/LO/2066). Results will be submitted for publication in peer-review journals, presented at conferences and inform the design of larger observational and experimental medicine studies assessing the role and management of post-TBI systemic inflammation.

## STRENGTHS AND LIMITATIONS OF THIS STUDY

⇒ Investigating the role of systemic inflammation is a novel approach to understanding potential mechanisms for the long-term effects of traumatic brain injury (TBI) such as neurodegeneration, and also provides potential targets for intervention.

⇒ We can fully exploit data from two large TBI studies with a relatively small amount of additional sample testing, with no further data collection required.

⇒ This is a unique cohort, comprising multimodal patient assessments (neuroimaging, clinical information and outcomes, blood biomarkers) across multiple time points spanning the acute to chronic period.

⇒ This study will only assess cytokines and inflammatory proteins, which is only one aspect of immune response.

## INTRODUCTION

Proinflammatory cytokine cascades, from maladaptive activation of microglia, are thought to have a key role in protein misfolding, misaggregation and subsequent neurodegeneration.[1] Animal studies demonstrate that systemic inflammation is signalled to the brain, resulting in transient changes in behaviour and brain metabolism, so-called 'sickness behaviour'.[2] However, it is increasingly recognised that this signalling can also result in the sustained and aggressive microglial activation associated with widespread neurodegeneration. Microglia may be 'primed' by established neurodegeneration, or in older brains, to respond abnormally to systemic inflammatory signals, which may explain apparent disease progression in patients with dementia after an infection.[3] Furthermore, some systemic inflammatory signals may be sufficiently chronic or severe

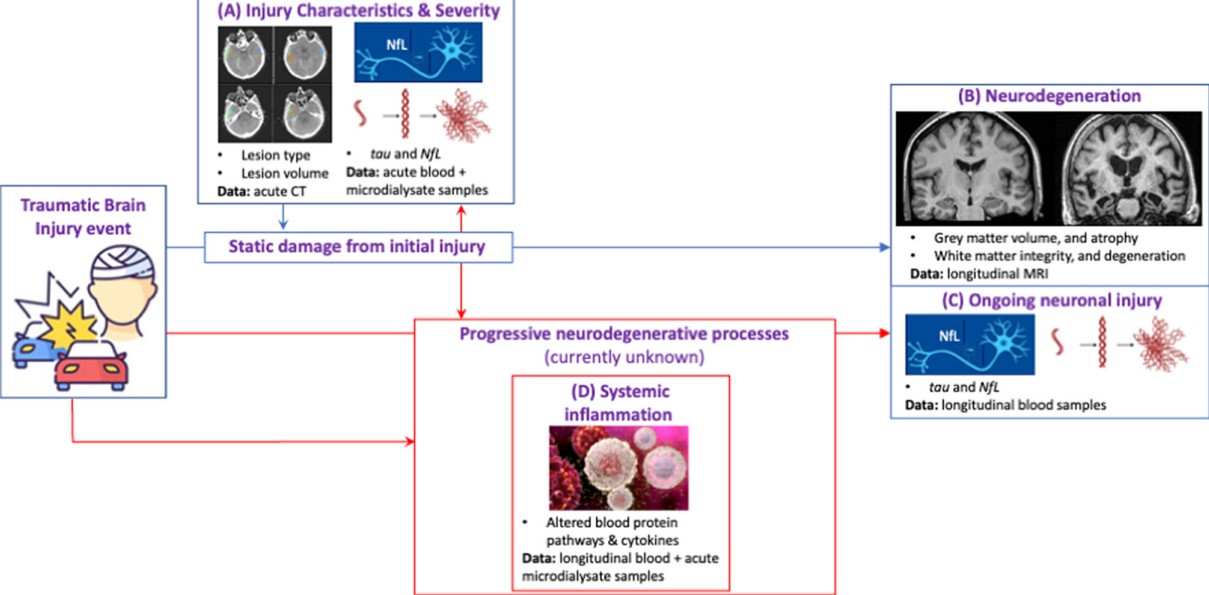

**Figure 1** Schematic of neurodegeneration after traumatic brain injury (TBI). Neuronal loss after TBI is a combination of the fixed damage from the initial injury, coupled with ongoing loss from progressive neurodegeneration. The static damage from the initial injury can be assessed by blood biomarkers and acute CT (A). Neurodegeneration can be visualised in vivo by changes in grey and white matter structure on longitudinal MRI (B) and through chronically elevated neuronal injury biomarkers (C). This project will investigate the relationship between injury and systemic inflammation in the acute and chronic phase (D) and the extent to which it contributes to ongoing neurodegeneration.

and trigger abnormal, ongoing neuroinflammation and resulting neurodegeneration.[4]

Traumatic brain injury (TBI) is an important environmental risk factor for neurodegenerative disease.[5] Both repeated mild and single severe injuries lead to chronic tau deposition and white matter degeneration.[6–8] However, the mechanisms by which TBI leads to neurodegeneration are not well elucidated, particularly those mechanisms causing progressive neurodegeneration, rather than the static damage from initial injury. Neurotrauma may cause direct peripheral immune infiltration through blood–brain barrier disruption,[9] and raised blood proinflammatory cytokines have been demonstrated after moderate-to-severe TBI.[10] Blood–brain barrier breakdown after trauma may cause centrally derived cytokines to leak into the wider circulation, but changes in serum cytokines after TBI are not entirely attributable to this. There is evidence, largely from animal studies, that cytokine production from peripheral immune organs, particularly the spleen and thymus, contributes to a systemic inflammatory response.[11] Change in sympathetic tone on lymphoid organs after TBI may also be a factor resulting in changes in peripheral cytokine profile.[12] However, no previous work has investigated whether this post-TBI systemic inflammation is linked to neurodegeneration. This project will examine the relationship between post-TBI systemic inflammation and ongoing neurodegeneration (figure 1).

TBI is highly prevalent, with 400+ daily UK hospital admissions, increasingly survivable, and especially increasing in older adults, itself a growing demographic.[13 14] Therefore, there is a pressing need to understand how it causes

neurodegeneration, and develop targeted interventions. We are investigating systemic as a potential mediator of neurodegeneration after TBI. This novel approach has the potential to identify novel targets for intervention, to improve long-term outcomes from TBI and mitigate against later neurodegeneration.

## METHODS AND ANALYSIS
### The aim
This project aims to evaluate extent of systemic inflammation and its contribution to ongoing neurodegeneration after TBI.

### The objectives
1. Investigate systemic inflammation in the acute and chronic post-TBI period, by characterising pattern and timeline of cytokine levels.
   - Compare the response in patients with TBI with non-TBI trauma and healthy controls.
2. Explore the relationship between acute cytokine levels in the blood and the brain extracellular fluid.
3. Determine the relationship between initial injury characteristics, as assessed by acute CT and initial clinical characteristics, and systemic inflammation in the acute and chronic postinjury period.
4. Investigate how systemic inflammation contributes to ongoing neurodegeneration, as assessed by longitudinal MRI and neuronal biomarkers NfL, GFAP and tau.

### Study design
This study combines data from two existing TBI patient longitudinal and prospective cohort

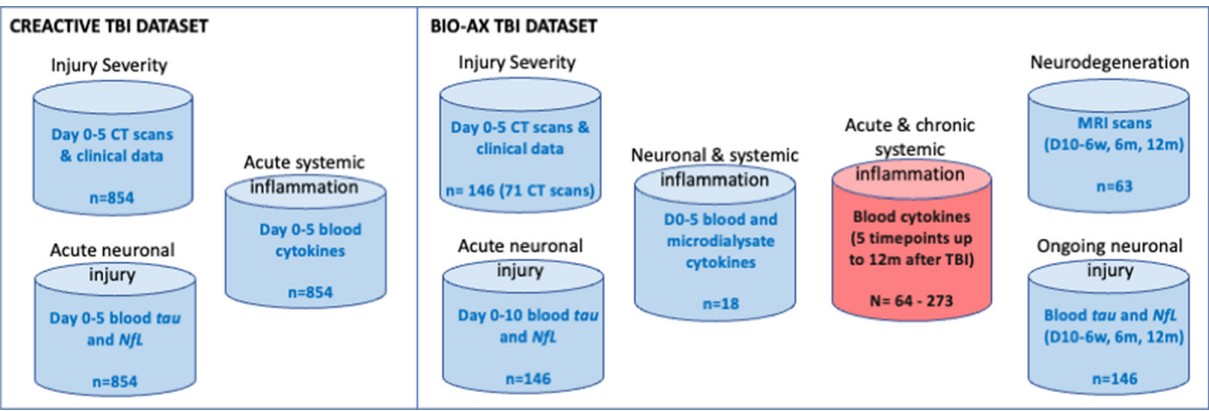

**Figure 2** Schematic showing TBI patient data available and to be acquired for this study. Blue indicates the data are already acquired, and red indicates data to be acquired during this study. D0–5=period from day of injury to 5 days after injury. D0–10=period from day of injury to 10 days after injury. D10–6w=period covering 10 days to 6 weeks after injury. 6m/12m=6 months/12 months. TBI, traumatic brain injury.

studies—CREACTIVE[15] (trial registration number: NCT02004080) and BIO-AX-TBI[16] (trial registration number: NCT03534154 (figure 2). In brief, CREACTIVE was a multicentre study, which recruited adults with TBI admitted into the intensive care unit (ICU) and prospectively followed them up at 6 months postinjury. Data collected included: blood samples at two time points (day 0 and day 5 of admission to ICU), initial CT brain scan, clinical information pertaining to injury and ICU events and 6-month functional outcome. The BIO-AX-TBI study was a multicentre study which recruited adult patients with moderate-to-severe TBI (Mayo classification)[17] to undergo prospective longitudinal multimodal assessments. Data collected included: blood samples at up to five time points between day 0 and 12 months after injury, initial CT brain scans, advanced MRI brain scans (including high-resolution anatomical T1 scan and diffusion tensor imaging) at up to three time points (between 6 weeks and 12 months after injury), clinical information pertaining to injury and 6-month and 12-month functional outcome. A subset of patients with TBI in BIO-AX-TBI also had paired acute blood and microdialysate levels collected. BIO-AX-TBI also collected blood samples from healthy and hospitalised non-TBI trauma control participants, and 3T advanced MRI brain scans from healthy controls only.

All blood samples have already been tested for tau and NfL using ultrasensitive Single molecule array (Simoa) technology. Inflammatory cytokines have been assessed in blood samples from CREACTIVE using the Olink Target 96 Inflammation panel (Olink Proteomics AB, Uppsala, Sweden) according to the manufacturer's instructions. All MRI brain scans have been analysed to extract measures of white matter integrity and grey matter volume.[18] During this study, we will assess inflammatory cytokine levels in the longitudinal blood samples collected from all BIO-AX participants (both control and TBI), as well as the paired microdialysate and blood samples (figure 2).

### Study setting
Both CREACTIVE and BIO-AX-TBI studies were prospective European multicentre international cohort studies.

### Participants and recruitment
Adults with TBI were recruited for CREACTIVE and BIO-AX-TBI. CREACTIVE recruited patients with TBI at the point of admission to ICU in multiple centres in Italy, Slovenia, Hungary, Poland, Cyprus, Israel and Greece.[19] These were all centres with familiarity with neurotrauma. Although TBI of any severity could be recruited, with some patients with mild TBI being admitted to ICU for other causes, the majority of cases were moderate-to-severe TBI. BIO-AX recruited any moderate-to-severe[17] patients with TBI from multiple centres in UK, Italy, Switzerland, Slovenia.[16] There will be no additional recruitment or sample collection.

### Sample size calculation
The scale of data obtained from this project will be unique in terms of the combination of the number of peripheral cytokines assayed, the size of TBI cohort and the number of time points. Our analysis approach for this discovery work will necessarily have exploratory aspects, from which specific hypotheses can be generated and tested, by us and future research projects.

Sample size for this study has been determined by the availability of data from CREACTIVE and BIO-AX-TBI (figure 2). Analysis of neuronal biomarkers and MRI scans has demonstrated adequate powering for this type of analysis.[18] A previous study demonstrated statistically significant differences in tested cytokine between patients with TBI at 3 months compared with healthy controls, and significant associations between cytokine levels and functional outcome.[10] This study had data from 70 to 80 patients for the analyses, so our study will likely be adequately powered to perform the intended analyses.

## Outline of study visits and procedures

No further study visits are required as we will test samples which have already been collected and are currently in deep freeze storage (at −80°C). Sample collection and processing was harmonised across CREACTIVE and BIO-AX-TBI. Blood samples were rested for 30 min after collection, then spun down (2000–2500 g for 10–20 min), and then placed into deep freeze storage at −80°C until thawed for testing. Multiple samples per participant were banked from each blood draw. Samples to be tested for the main analyses in this study will be those that have not undergone any previous freeze-thaw cycles. A subset of samples which have undergone one previous freeze-thaw cycle will be tested to specifically investigate the impact of freeze-thaw cycles on cytokine levels. We will test for inflammatory cytokines in blood and microdialysate samples already collected and stored as part of the BIO-AX-TBI study.

We will use the OLINK testing platform[20] based at the UK DRI Fluid Biomarker lab. Technical and internal controls are used to assess and maintain quality of data. OLINK samples are run in single. OLINK technology has a built in quality control that enables performance monitoring in terms of reproducibility and coefficient of variation, with strict requirements on performance. OLINK assays are designed and validated to be physiologically relevant in plasma/serum. Further technical detail is available on the OLINK website (https://olink.com/our-platform/assay-validation/). It is still possible for data to fall below lower limit of detection (LOD), and we will follow OLINK guidance on how to handle this data (https://olink.com/faq/how-is-the-limit-of-detection-lod-estimated-and-handled/). In brief, we plan to use the actual data value, but will also assess for any significant between-group difference in the proportion of samples below LOD for the given protein of interest.

We will perform the following tests:

► Testing of all BIO-AX-TBI samples with the OLINK Target 96 Inflammation panel—this is the same panel used on the already tested CREACTIVE samples, and assesses levels of 92 proteins, including common inflammatory cytokines and proteins. A small number of samples, capturing the dynamic range within the CREACTIVE data, will be tested at the same time as the BIO-AX-TBI samples to facilitate bridge normalisation across the entire data set. This enables the direct comparison of protein levels across samples run at different times on different reagent batches, removing any batch affects and aligning the data to a common reference point. The resulting data are on a concrete, relative scale (NPX), with a difference of 1 NPX equating to a doubling in protein concentration. Values for each protein can be compared across experimental groups and time points within any one protein.

► Testing of a subset of BIO-AX-TBI and the already tested CREACTIVE samples on the OLINK Target 48 Cytokine panel. This panel provides both normalised and absolute values for 45 proteins associated with inflammation. The panel shows good correlation at validation with overlapping proteins on OLINK Target 96 Inflammation. We aim to use this subset of absolute data on OLINK Target 48 Cytokine panel to create a model to translate the relative readout from OLINK Target 96 Inflammation to an absolute data set. The success of this will be tested using validation samples run on both OLINK Target 96 Inflammation and OLINK Target 48 Cytokine panel.

► Test the paired microdialysate and blood samples on the OLINK Target 48 Cytokine Panel to examine the relationship between CNS and peripheral cytokine levels.

► Test a subset of previously defrosted plasma samples (one previous freeze-thaw cycle) to ascertain whether the freeze-thaw process affects cytokine levels. This has implications for biomarker testing for future studies, as it means that utility of samples could be extended, maximising the use derived from precious human samples.

## Outcomes

### Injury severity and extent

This will be assessed through routinely collected clinical data, for example, admission Glasgow Coma Scale and characteristics of the initial CT brain scan. The CT brain scans have Marshall grading,[21] indicating type and extent of injury, and a large subset of the scans have also been analysed with a semiautomated pipeline that extract lesion type and volume.[22 23]

### Blood neuronal biomarkers

In the combined CREACTIVE and BIO-AX-TBI dataset, blood levels of *tau, NFL, GFAP, UCHL1, S100B* have been tested in patients with TBI at multiple time points: 0–10 days (2 samples), 10 days to 6 weeks, 6 months and 12 months after injury. Blood neuronal biomarker levels are also available in from the healthy controls and acutely from the non-TBI trauma controls. GFAP has an extremely early peak after TBI and is thought to be related to severity of injury.[24] *NFL* levels peak at 3 weeks after moderate-to-severe TBI and likely reflects the initial damage whereas, in the chronic period, both *NFL* and *tau* are markers of ongoing neuronal injury or loss.[18 25 26]

### Systemic inflammation

Inflammatory cytokine and protein levels in blood will be assessed at multiple time points after TBI to enable characterisation of the systemic inflammatory response over time.

► Acute time points: day 0 and day 5 (CREACTIVE) and two samples between days 0 and 10 (BIO-AX-TBI).

► Subacute time point—day 10 to 6 weeks (BIO-AX-TBI).

► Two chronic time points—6 and 12 months (BIO-AX-TBI).

Concentrations will also be assessed in blood samples from the healthy and non-TBI trauma controls, and the

paired microdialysate-blood samples obtained in the acute period after TBI.

Prior studies looking at inflammatory proteins after TBI have shown prolonged postinjury elevations in IL6, IL-1β, TNF-α, IL8, IL10, VEGF, as well as associations between these cytokines and functional outcomes.[10 27–30]

Some routinely collected clinical data will also be available for BIO-AX-TBI samples, for example, blood C reactive protein level and white cell count (WCC). For CREACTIVE samples, the SAPS II categorisation of the WCC is available.

### Neurodegeneration

TBI leads to ongoing neurodegeneration, which can be assessed via changes in white matter integrity and grey matter volume in MRI.[31–33] The extent of initial white matter injury also predicts extent of later ongoing neurodegeneration.[33 34] White matter integrity, grey and white matter volume, and the change in these measures over time will be used as markers of neurodegeneration. These measures have been extracted from the longitudinal MRI scans (10 days to 6 weeks, 6 and 12 months) collected in the BIO-AX-TBI study.[18] NFL and tau levels in the chronic period will also be used as indicators of neurodegeneration.

### Clinical outcomes

The Glasgow Outcome Scale Extended will be used to assess global functional outcome postinjury and is available at 6 months from CREACTIVE and BIO-AX-TBI patients, and at 12 months from BIO-AX-TBI patients. More detailed outcome measures are available in the BIO-AX-TBI patients, including neuropsychological tests and symptom questionnaires.[16]

## Data handling
### Missing values

Missing data can lead to biased or less precise estimates. Our a priori plan is to use multiple imputation to handle missing data. This involves creating several sets of imputed data, each with a plausible value for the missing data, and then analysing each of the completed data sets using the same statistical model. The results from the multiple analyses are then combined to obtain an overall estimate, with SEs that reflect both within-imputation and between-imputation variability. Before applying multiple imputation, we will carefully examine the patterns of missing data and the reasons for missingness. We will report the missing data patterns and, where known, reasons for missingness. Other methods of handling missing data may be used if a specific analysis or characteristics of the data make multiple imputation inappropriate.

### Significance threshold

As a general rule, we will set the alpha value at 0.05, as is common in the field. However, in some cases, a more conservative alpha value may be appropriate to reduce the risk of false positive results, for example, to account for multiple comparisons. We will also consider the potential impact of using different alpha values on the interpretation of our results. We will discuss our chosen alpha value in the methods section of any results papers. We also note that, given the large sample and range of analyses planned, effect size is likely to be more meaningful than significance value.

## Data analysis

Data analysis will be aimed at addressing the key aims of this project. Some analyses will use pooled CREACTIVE and BIO-AX-TBI, while other analyses will be limited to the BIO-AX-TBI data. Analyses will be performed using Python and R scripts developed in-house.

### Analysis 1. *Characterising pattern and timeline of systemic inflammation after TBI, as compared with non-TBI trauma*

A cluster analysis approach will be used to characterise the pattern of blood cytokine level response in the acute and chronic postinjury period, and curve-fitting approaches (eg, local polynomial regression) will be used to approximate the trajectory. Clusters will be calculated separately for the acute (up to 10 days) and chronic periods (6 months or more), as well as the overall time. This separate assessment is important because the cytokine profiles during acute and chronic periods may well have different drivers and different consequences. An iterative subsampling method (subsampling from the large TBI group to compare against the smaller healthy and non-TBI trauma control groups) will be used to identify 'discriminatory' cytokines (ie, those which most robustly distinguish between TBI and the other two groups. These analyses will be carried out on pooled CREACTIVE and BIO-AX-TBI data, comprising samples from 918 up to 1127 TBI participants across 5 time points, and 102 healthy and 24 non-TBI trauma controls (1 time point).

### Analysis 2. *Exploring the relationship between acute cytokines in the blood and brain extracellular fluid*

We will test whether systemic and intracranial cytokine levels are correlated using simple correlation analyses on the paired microdialysate and blood data (from 18 participants). The focus will be specifically on the 'discriminatory' cytokines. Bonferroni correction will be used to correct for multiple comparisons (based on number of cytokines investigated in the correlation analyses).

### Analysis 3. *Investigating the relationship between initial injury and systemic inflammation in both the acute (days 0–10) and chronic (days 10–12 months) periods*

Pooled CREACTIVE and BIO-AX-TBI data will be used to assess the relationship in the acute period (defined as <10 days), with data from 1000 TBI participants. Only BIO-AX-TBI data will be used to assess the relationship in the chronic period, with data from up to 146 TBI participants. Patients with TBI will be grouped based on cluster analyses of the cytokine response during the acute and chronic periods (see Analysis 1.). Linear mixed effect model analysis will be used to identify the injury characteristics most predictive of group membership. Injury characteristics

will include clinical and CT data as described above, as well as acute blood tau and NFL levels. Acutely, blood tau and NFL likely reflect extent of neuronal damage from the initial injury and are also predictive of functional outcome.[18 35] Potential confounders will be accounted for, for example, age.

### Analysis 4. *Investigating relationship between systemic inflammation and progressive neurodegeneration*

Only BIO-AX-TBI data, from 63 participants, will be included in this analysis. The change in grey matter volume on MRI over the three time points, using the Jacobian determinant,[36] and blood NfL and tau in the chronic period (6 and 12 months) will be used as measures of neurodegeneration. NfL may be a particularly sensitive marker of ongoing neuronal injury occurring in the context of a progressive neurodegenerative process, and in animal models coincides with onset of neurodegenerative proteinopathy.[25 26]

Linear Mixed Model (LMM) analyses will be used to assess how well measures of neurodegeneration can be predictive by systemic inflammation, while accounting for injury severity markers. The predictive value of both cytokine clusters and individual 'discriminatory' cytokines will be assessed. The latter analysis will aid identification of specific cytokines which might be targets for future intervention. LMM enables modelling of repeated measures over time, and to account for within-subject correlation. This reduces the number of statistical tests needed, which reduces the need to correct for multiple comparisons. We will also use Bonferroni correction or another appropriate method to adjust for multiple comparisons in this analysis.

Mediation analysis will be used to explore the extent to which relationships between injury characteristics and neurodegeneration is mediated by specific cytokines or cytokine clusters. We will construct directed acyclic graphs (DAG) to help us identify the different potential causal pathways between variables of interest and guide our mediation analysis. We will construct several DAGs to represent different theoretical models that match different temporal time points, with the variables of interest positioned in the appropriate order based on prior knowledge and theoretical considerations. The DAG will allow us to identify the potential direct and indirect effects of the variables of interest on the outcome, and to test the significance of the indirect effects using appropriate statistical methods. Figure 3 is a DAG illustrating our key hypotheses: (1) that injury severity and extent directly influence neurodegeneration and (2) cytokine levels in the chronic period separately mediate the relationship between injury severity and extent and neurodegeneration. In this case, the mediation analysis

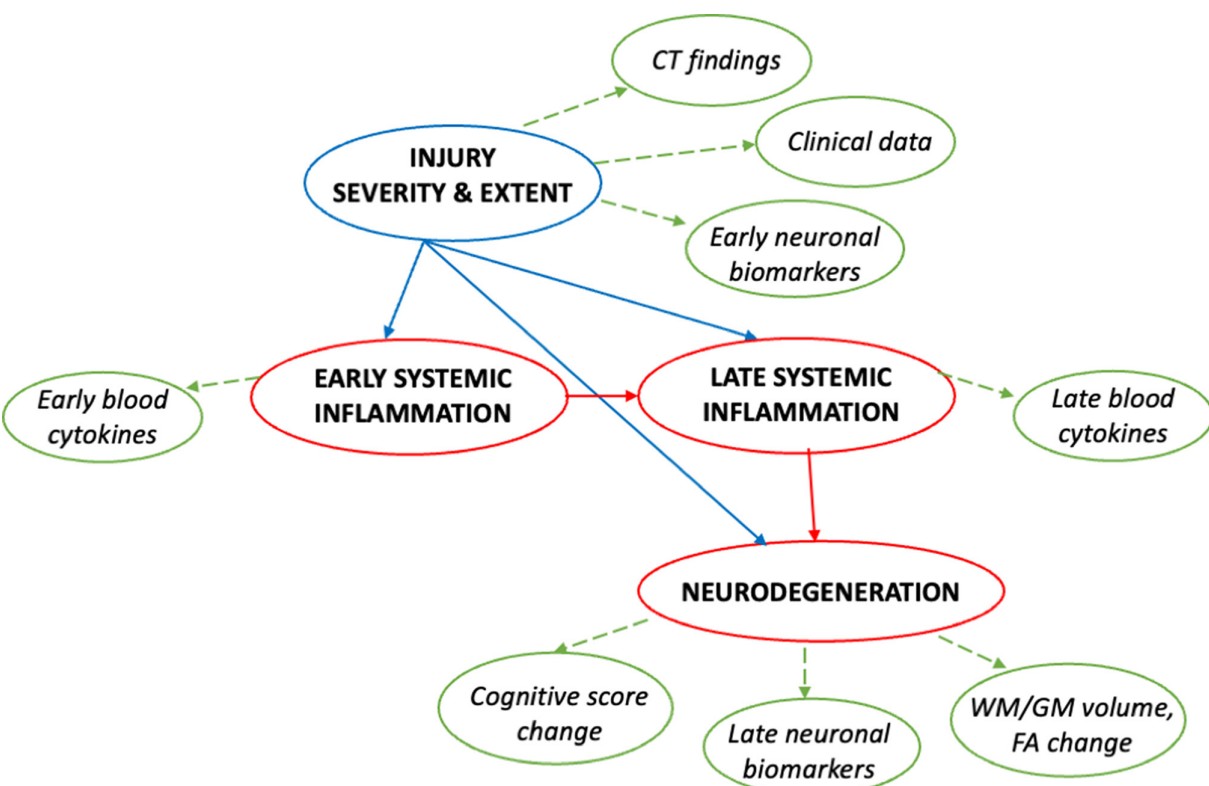

**Figure 3** In this directed acyclic graph, injury severity and extent are hypothesised to cause both early (up to 6 weeks) and late (6+ months) systemic inflammation, as well as directly influence neurodegeneration. Additionally, late systemic inflammation is hypothesised to partly mediate the relationship between injury severity and extent and neurodegeneration. Red denotes processes, blue indicates events and green denotes outcome measures. FA, fractional anisotropy; GM, grey matter; WM, white matter.

will focus on the extent to which specific cytokines or cytokine clusters in the chronic period mediate the relationship between injury severity and extent and neurodegeneration (figure 3). Further DAGs will be constructed based on findings from above analyses. In addition, we will conduct sensitivity analyses to assess the robustness of our findings to potential violations of the DAG assumptions, such as unmeasured confounding. We will report our DAG and sensitivity analysis results alongside main findings in any results papers.

## PATIENT AND PUBLIC INVOLVEMENT

There was no specific patient and public involvement in the conceptualisation and design of this study. However, we hold regular outreach events at Imperial College London including where our research programme is discussed with participants and interest public. Feedback is sought and incorporated into the design of our work. Data from this study will be disseminated widely, including in presentations at outreach events where patients, families and other interested lay members are present.

## ETHICS AND DISSEMINATION

Ethical approval for this study has been granted by the London—Camberwell St Giles Research Ethics Committee (REC number: 17/LO/2066). Approval for the sharing of data already collected has been separately granted as part of the ethical approvals for the BIO-AX-TBI and CREACTIVE studies.[15 16]

Results from cytokine tests, processed CT and MRI scans, will be stored on the Research Data Repository managed by Imperial College. These results will be stored under non-identifying project IDs. The results from cytokine tests will also be entered into the secure online database management system REDCap (with which Imperial College hold an institutional license), and these will be stored alongside identifying information. Finalised versions of codes will be uploaded onto GitHub and also stored on the Imperial College Research Data Repository and in project folders on secure cloud-based servers operated through Imperial College London. LML will provide project management and oversight in conjunction with the leads of the CREACTIVE (GB) and BIO-AX-TBI studies (DJS).

## TRIAL STATUS

We are in the process of collating all additional samples to be tested in this study. We anticipate testing to take place at the end of 2022/start of 2023.

**Author affiliations**
[1]Brain Sciences, Imperial College, London, UK
[2]UKDRI Centre for Care Research & Technology, London, UK
[3]Department of Neurodegenerative Disease, UCL Queen Square Institute of Neurology, London, UK
[4]UKDRI at UCL, London, UK
[5]IRCCS-"Mario Negri" Institute for Pharmacological Research, Ranica, Bergamo, Italy
[6]BioMedIA Group, Department of Computing, Imperial College, London, UK
[7]Istituto Di Ricerche Farmacologiche Mario Negri, Ranica, Italy
[8]DRI Centre for Care Research and Technology, London, UK
[9]Mario Negri Institute for Pharmacological Research, Milan, Italy
[10]Cardiovascular Medicine, Mario Negri Institute for Pharmacological Research, Milan, Italy
[11]Clinical Dpt of Anaesthesiology and Intensive Therapy, University Medical Center, Ljubljana, Slovenia
[12]Faculty of Medicine, University of Ljubljana, Ljubljana, Slovenia
[13]Department of Anesthesia and Intensive Care, Santa Chiara Hospital, Trento, Italy
[14]Department of Psychiatry and Neurochemistry, Institute of Neuroscience and Physiology, Sahlgrenska Academy, University of Gothenburg, Gothenburg, Sweden
[15]Clinical Neurochemistry Laboratory, Sahlgrenska University Hospital, Mölndal, Sweden
[16]Public Health, Laboratory of Clinical Epidemiology, IRCCS-"Mario Negri" Institute for Pharmacological Research, Ranica, Italy
[17]Division of Brain Sciences, Imperial College, London, UK

**Acknowledgements** We thank Dr Janet Kenyon (Olink Proteomics AB) for her input with biomarker panel selection and testing strategy.

**Contributors** LL is the project principal investigator and has project oversight along with DJS and GB. AH, ES, GN, GB, MR, BG, HZ and DJS helped write the successful grant application to obtain funding for the project. AH and HZ tested the samples and helped design the sample testing strategy. MR, BG, KAZ, NSNG and FM helped design the neuroimaging analysis strategy. EG, KAZ, NSNG, FM, DN, PG and SM collected data to be analysed and helped formulate hypotheses and design the overall data analysis strategy. DN manages sample storage and transfer. ES and GN helped design the overall statistical testing. LL wrote the first draft of the manuscript. All authors reviewed and edited the manuscript.

**Funding** This project is funded by a UK DRI Pilot Grant awarded to LL. LL and NSNG are funded by the NIHR with Clinical Lectureship fellowship grants. HZ is a Wallenberg Scholar supported by grants from the Swedish Research Council (#2018-02532), the European Union's Horizon Europe research and innovation programme under grant agreement No 101053962, Swedish State Support for Clinical Research (#ALFGBG71320), the Alzheimer Drug Discovery Foundation (ADDF), USA (#201809-2016862), the AD Strategic Fund and the Alzheimer's Association (#ADSF-21-831376-C, #ADSF-21-831381-C, and #ADSF-21-831377-C), the Bluefield Project, the Olav Thon Foundation, the ErlingPersson Family Foundation, Stiftelsen för Gamla Tjänarinnor, Hjärnfonden, Sweden (#FO2022-0270), the European Union's Horizon 2020 research and innovation programme under the Marie Skłodowska-Curie grant agreement No 860197 (MIRIADE), the European Union Joint Programme-Neurodegenerative Disease Research (JPND2021-00694), and the UK Dementia Research Institute at UCL (UKDRI-1003).

**Competing interests** HZ has served at scientific advisory boards and/or as a consultant for Abbvie, Acumen, Alector, ALZPath, Annexon, Apellis, Artery Therapeutics, AZTherapies, CogRx, Denali, Eisai, Nervgen, Novo Nordisk, Passage Bio, Pinteon Therapeutics, Red Abbey Labs, reMYND, Roche, Samumed, Siemens Healthineers, Triplet Therapeutics and Wave, has given lectures in symposia sponsored by Cellectricon, Fujirebio, Alzecure, Biogen and Roche, and is a cofounder of Brain Biomarker Solutions in Gothenburg AB (BBS), which is a part of the GU Ventures Incubator Programme (outside submitted work).

**Patient and public involvement** Patients and/or the public were not involved in the design, or conduct, or reporting, or dissemination plans of this research.

**Patient consent for publication** Not applicable.

**Provenance and peer review** Not commissioned; externally peer reviewed.

**Open access** This is an open access article distributed in accordance with the Creative Commons Attribution 4.0 Unported (CC BY 4.0) license, which permits others to copy, redistribute, remix, transform and build upon this work for any purpose, provided the original work is properly cited, a link to the licence is given, and indication of whether changes were made. See: https://creativecommons.org/licenses/by/4.0/.

**ORCID iDs**
Lucia M Li http://orcid.org/0000-0001-7339-4854

Elena Garbero http://orcid.org/0000-0003-4902-0144
Neil S N Graham http://orcid.org/0000-0002-0183-3368

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
