## [Reviewer comments · BMJ Open]

ARTICLE DETAILS

TITLE (PROVISIONAL)	Investigating the Characteristics and Correlates of Systemic Inflammation after Traumatic Brain Injury: the TBI-BraINFLAMM study
AUTHORS	Li, Lucia; Heslegrave, Amanda; Soreq, Eyal; Nattino, Giovanni; Rosnati, Margherita; Garbero, Elena; Zimmerman, Karl; Graham, Neil; Moro, Federico; Novelli, Deborah; Gradisek, Primoz; Magnoni, Sandra; Glocker, Ben; Zetterberg, Henrik; Bertolini, Guido; Sharp, David

VERSION 1 – REVIEW

REVIEWER	Mader, Marius Stanford University School of Medicine
REVIEW RETURNED	21-Nov-2022

GENERAL COMMENTS	The authors report a study protocol for the TBI-BraINFLAMM study. The TBI-BraINFLAMM is the combination of already existing data/samples of two prospective TBI studies (CREACTIVE and BIO-AX-TBI). The dataset will provide clinical data, imaging data and blood samples and pre-existing analysis data. The authors plan to run additional inflammatory cytokine analyses on the blood and microdialysate samples of the BIO-AX-TBI study. The protocol is well written and main relevant information is provided. I have no major issues. Can the authors add additional information on sample management. E.g. were samples collected and handled similarly between the two initial studies. Number of freeze-thaw cycles? Moreover, what clinical data will be available - for your study question laboratory data like immune cell counts etc would be of interest? I wish the authors success with their project!
---

REVIEWER	Meier, Timothy Medical College of Wisconsin, Neurosurgery
REVIEW RETURNED	23-Jan-2023

GENERAL COMMENTS	This is a protocol for an interesting study investigating systemic inflammation following TBI and its association with injury several and markers neurodegeneration. The justification and objectives are clearly laid out. There are some aspects of the protocol that need to be clarified or expanded. This information will be critical for tracking whether ultimate analyses follow this specified protocol. The authors should consider doing a proper power analysis for each of the described analyses. Because this is a combination of existing/ongoing studies, the sample size is set. Thus, power
---

	analyses can be used to demonstrate the effect size that this study is powered to observed. Overall, the additional blood assays are well described. However, details are missing. Are samples to be run in duplicate? If so, how will samples with high CV be handled? Finally, is there a priori plan for handling biomarkers that are outside limits of detection? Can the authors clarify what is meant by normalizing the output values to allow cytokine comparisons between timepoints and groups? Do the authors foresee log transforming the biomarkers prior to statistical analysis? The number of participants expected to be included in each analysis is not clear and could be clarified. In the statistical approach, what alpha will be considered for each analysis? For analysis 4 in particular, will there be any consideration for accounting for multiple comparisons? Is there a priori plan for handling missing data in analyses? The description of the mediation analyses can be expanded. Directionality and temporality are critical in mediation analysis. The authors should consider a directed acyclic graph to determine the position of the variables of interest in their model based on a priori and theoretical considerations.
--	--

REVIEWER	McDonald, Stuart Monash University
REVIEW RETURNED	24-Jan-2023

GENERAL COMMENTS	This is an important and well described study that leverages existing data and samples generated from two large prospective TBI studies. The protocol is thorough and appropriate to investigate the primary aims of the study. There are no apparent flaws that will prevent sound interpretation of the data. One minor suggestion is for the introduction. This section provides the rationale for how systemic inflammation might be a mediator of neurodegeneration. How TBI can lead to systemic inflammation is not raised. The authors might consider outlining what is known or hypothesised about the primary cellular sources of raised blood cytokines after TBI. Congratulations on nice protocol and exciting study. I look forward to seeing the results.
---

VERSION 1 – AUTHOR RESPONSE

Reviewer: 1

Dr. Marius Mader, Stanford University School of Medicine

Comments to the Author:

The authors report a study protocol for the TBI-BraINFLAMM study. The TBI-BraINFLAMM is the combination of already existing data/samples of two prospective TBI studies (CREACTIVE and BIO-AX-TBI). The dataset will provide clinical data, imaging data and blood samples and pre-existing analysis data. The authors plan to run additional inflammatory cytokine analyses on the blood and microdialysate samples of the BIO-AX-TBI study.

The protocol is well written and main relevant information is provided. I have no major issues. Can the authors add additional information on sample management. E.g. were samples collected and handled similarly between the two initial studies. Number of freeze-thaw cycles? Moreover, what clinical data will be available - for your study question laboratory data like immune cell counts etc would be of interest?

I wish the authors success with their project!

Thank you to Dr Mader for his positive comments. We are pleased to provide clarification within the manuscript on the important points he has raised, as below:

Page 4, Outline of study visits & procedures:

“Sample collection and processing was harmonised across CREATIVe and BIO-AX-TBI. Blood samples were rested for 30mins after collection, then spun down (2000-2500g for 10-20mins), and then placed into deep freeze storage at -80°C until thawed for testing. Multiple samples per participant were banked from each blood draw. Samples to be tested for the main analyses in this study will be those that have not undergone any previous freeze-thaw cycles. A subset of samples which have undergone one previous freeze-thaw cycle will be tested to specifically investigate the impact of freeze-thaw cycles on cytokine levels.”

Page 6, Outcomes: Systemic Inflammation:

“Some routinely collected clinical data will also be available for BIO-AX-TBI samples, for example: blood C-reactive protein (CRP) level and white cell count (WCC). For CREATIVe samples, the SAPS II categorisation of the WCC is available.”

Reviewer: 2

Dr. Timothy Meier, Medical College of Wisconsin

Comments to the Author:

This is a protocol for an interesting study investigating systemic inflammation following TBI and its association with injury severity and markers neurodegeneration. The justification and objectives are clearly laid out. There are some aspects of the protocol that need to be clarified or expanded. This information will be critical for tracking whether ultimate analyses follow this specified protocol.

Thank you to Dr Meier for his detailed attention to our manuscript. We have made a number of substantial additions to the manuscript, including a section called “Data Handling” which addresses some of the general points raised by the reviewer.

The authors should consider doing a proper power analysis for each of the described analyses. Because this is a combination of existing/ongoing studies, the sample size is set. Thus, power analyses can be used to demonstrate the effect size that this study is powered to observed.

Thank you for your comment. We agree that power analyses are very important and helpful for hypothesis testing studies where a specific hypothesis is tested with a predetermined level of significance and power. However, we do not think it is possible to provide a meaningful power analysis for this protocol paper. This study is quite unique in the scale of the data to be collected (in terms of number of cytokines, size of TBI cohort and number of timepoints), so our analyses will initially be exploratory in nature. Our analyses also have some interdependence – later analyses exploring the relationship of peripheral cytokines with other biomarkers and outcomes will focus on the cytokines initially identified to be most discriminatory of TBI.

We have expanded the “Sample Size Calculation” section to clarify the above:

Page 4, Sample Size Calculation:

“The scale of data obtained from this project will be unique in terms of the combination of the number of peripheral cytokines assayed, the size of TBI cohort and the number of timepoints. Our analysis approach for this discovery work will necessarily have exploratory aspects, from which specific hypotheses can be generated and tested, by us and future research projects.

Sample size for this study has been determined by the availability of data from CREACTION and BIO-AX-TBI (FIGURE 2). Analysis of neuronal biomarkers and MRI scans has demonstrated adequate powering for this type of analysis¹⁸. A previous study demonstrated statistically significant differences in tested cytokine between TBI patients at 3 months compared to healthy controls, and significant associations between cytokine levels and functional outcome¹⁰. This study had data from 70-80 patients for the analyses, so our study will likely be adequately powered to perform the intended analyses.”

Overall, the additional blood assays are well described. However, details are missing. Are samples to be run in duplicate? If so, how will samples with high CV be handled? Finally, is there a priori plan for handling biomarkers that are outside limits of detection?

Thank you for these queries, which we clarify now. OLINK samples are run in single, as OLINK state that it is not necessary to run samples in duplicate. This is because OLINK technology has a built in QC that enables performance monitoring in terms of reproducibility and CV, with strict requirements on performance. Further technical detail is available on the OLINK website (<https://olink.com/our-platform/assay-validation/>).

With regards to limits of quantification (LOQ), OLINK assays are designed and validated to be physiologically relevant in plasma/serum. Therefore, these samples have at validation been shown to be within the LOQ. Further technical detail is available on the web page linked above. It is possible for data to fall below lower limit of detection and OLINK provide guidance on how to handle this data (<https://olink.com/faq/how-is-the-limit-of-detection-lod-estimated-and-handled/>). We plan to use the actual data value, assuming that there isn't a significant between-group difference in the proportion of samples below LOD for the given protein of interest.

We have added this explanation to the manuscript as below:

Page 4, Outline of study visits and procedures:

“OLINK samples are run in single. OLINK technology has a built in quality control that enables performance monitoring in terms of reproducibility and coefficient of variation, with strict requirements

on performance. OLINK assays are designed and validated to be physiologically relevant in plasma/serum. Further technical detail is available on the OLINK website (<https://olink.com/our-platform/assay-validation/>). It is still possible for data to fall below lower limit of detection (LOD), and we will follow OLINK guidance on how to handle this data (<https://olink.com/faq/how-is-the-limit-of-detection-lod-estimated-and-handled/>). In brief, we plan to use the actual data value, but will also assess for any significant between-group difference in the proportion of samples below LOD for the given protein of interest.”

Can the authors clarify what is meant by normalizing the output values to allow cytokine comparisons between timepoints and groups?

Thank you for pointing out that our explanations on this should be clearer. There are two aspects of normalisation which need to occur within the project. The first is to allow us to be able to combine results from the newly tested samples with the already tested samples. We will do this by testing samples from a small number of already-tested CREATIVE participants (multiple samples per participant were banked from participants at each blood draw) to form a ‘bridge’ cohort. The second is to enable us to obtain absolute concentrations of proteins for every single sample. The previously tested CREATIVE samples used the OLINK 96 panel. The resulting data is on a concrete, relative scale (NPX), with a difference of 1NPX equating to a doubling in protein concentration. In order to derive absolute (rather than relative) concentrations, we will test a subset of samples using the OLINK 48 panel, which provides data in both absolute concentrations and the relative NPX scale. We will then create a model to translate the relative readout from the OLINK 96 panel to absolute values.

We have amended the manuscript to provide further detail and clarification on this:

Page 5, “Outline of study visits & procedures:

- Testing of all BIO-AX-TBI samples with the OLINK® Target 96 Inflammation panel – this is the same panel used on the already tested CREATIVE samples, and assesses levels of 92 proteins, including common inflammatory cytokines and proteins. A small number of samples, capturing the dynamic range within the CREATIVE data, will be tested at the same time as the BIO-AX-TBI samples to facilitate bridge normalisation across the entire data set. This enables the direct comparison of protein levels across samples run at different times on different reagent batches, removing any batch effects and aligning the data to a common reference point. The resulting data is on a concrete, relative scale (NPX), with a difference of 1NPX equating to a doubling in protein concentration. Values for each protein can be compared across experimental groups and timepoints within any one protein.
- Testing of a subset of BIO-AX-TBI and the already tested CREATIVE samples on the OLINK® Target 48 Cytokine panel. This panel provides both normalised and absolute values for 45 proteins associated with inflammation. The panel shows good correlation at validation with overlapping proteins on OLINK® Target 96 Inflammation. We aim to use this subset of absolute data on OLINK® Target 48 Cytokine panel to create a model to translate the relative readout from OLINK® Target 96 Inflammation to an absolute data set. The success of this will be tested using validation samples run on both OLINK® Target 96 Inflammation & OLINK® Target 48 Cytokine panel.”

The number of participants expected to be included in each analysis is not clear and could be clarified.

Thank you to the reviewer for pointing out that this needs clarification. We provided a graphic to denote how many participants contribute data to each data type, and we refer to the data type in the manuscript text under the “Data Analysis” section. We have now added the numbers to the text as

well to make it clearer for the reader to understand how many participants are expected in each analysis.

Page 7-8, "Data Analysis:

1. *Characterising pattern and timeline of systemic inflammation after TBI, as compared to non-TBI trauma:*

These analyses will be carried out on pooled CREATIVE and BIO-AX-TBI data, comprising samples from 918 up to 1127 TBI participants across 5 timepoints, and 102 healthy and 24 non-TBI trauma controls.

2. *Exploring the relationship between acute cytokines in the blood and brain extracellular fluid: We will test whether systemic and intracranial cytokine levels are correlated using simple correlation analyses on the paired microdialysate and blood data (from 18 participants).*

3. *Investigating the relationship between initial injury and systemic inflammation in both the acute (Day 0 to 10) and chronic (Day 10 – 12 months) periods:*

Pooled CREATIVE and BIO-AX-TBI data will be used to assess the relationship in the acute period (defined as <10 days), with data from 1000 TBI participants. Only BIO-AX-TBI data will be used to assess the relationship in the chronic period, with data from up to 146 TBI participants.

4. *Investigating relationship between systemic inflammation and progressive neurodegeneration: Only BIO-AX-TBI data, from 63 participants, will be included in this analysis."*

Do the authors foresee log transforming the biomarkers prior to statistical analysis?

Thank you for this question. The decision to log-transform biomarkers prior to statistical analysis will depend on the specific characteristics of the data and the goals of the specific analysis.

Once we have the data, we will carefully evaluate the distribution of each biomarker to determine whether a log-transformation is necessary or appropriate. If the data is heavily skewed, we may consider applying a log-transformation to normalize the distribution and improve the accuracy of the statistical analysis. However, if the data is normally distributed or only slightly skewed, a log-transformation may not be necessary or appropriate.

We will also consider the potential effects of log-transforming the biomarker data on the interpretation of our results. While a log-transformation can improve the accuracy of statistical analysis, it can also make the results more difficult to interpret or explain to a general audience. Therefore, we will carefully consider the potential benefits and drawbacks of log-transforming the data before making a decision.

In future manuscripts, the decision to perform log transformation or not will be discussed in 'Methods' and 'Results' sections.

Is there a priori plan for handling missing data in analyses?

Thank you for your question. We recognise that missing data is a common issue which can lead to biased or less precise estimates if not handled appropriately. Therefore, we do have an *a priori* plan for handling missing data.

Our *a priori* plan is to use multiple imputation to handle missing data. Multiple imputation involves creating several sets of imputed data, each with a plausible value for the missing data, and analyzing each of the completed data sets using the same statistical model. The results from the

multiple analyses are then combined to obtain an overall estimate, with standard errors that reflect both within- and between-imputation variability.

Before applying multiple imputation, we will carefully examine the patterns of missing data and the reasons for missingness. We will report the missing data patterns and reasons for missingness in the results section of our manuscript.

We note that, in addition to multiple imputation, we may also consider other methods such as maximum likelihood or full-information maximum likelihood, depending on the specific analysis and the characteristics of the data.

We have added a section to the 'Data Analysis' section to clarify the above:

Page 6-7, Data Handling:

"Handling missing values:

Missing data can lead to biased or less precise estimates. Our a priori plan is to use multiple imputation to handle missing data. This involves creating several sets of imputed data, each with a plausible value for the missing data, and then analysing each of the completed data sets using the same statistical model. The results from the multiple analyses are then combined to obtain an overall estimate, with standard errors that reflect both within- and between-imputation variability. Before applying multiple imputation, we will carefully examine the patterns of missing data and the reasons for missingness. We will also assess whether the data is missing completely at random, missing at random, or missing not at random, and we will adjust our imputation model accordingly. We will report the missing data patterns and, where known, reasons for missingness. Other methods of handling missing data may be used if a specific analysis or characteristics of the data make multiple imputation inappropriate."

In the statistical approach, what alpha will be considered for each analysis?

Thank you for your question. We plan to carefully consider the appropriate alpha value for each analysis based on the specific research question being addressed, the type of statistical test being used, and any relevant literature in the field. As a general rule, we will set the alpha value at 0.05, as is common in the field. However, in some cases, a more conservative alpha value may be appropriate to reduce the risk of false positive results, for example, to account for multiple comparisons. We will also consider the potential impact of using different alpha values on the interpretation of our results. We will discuss our chosen alpha value in the methods section of any results papers. We also note that, given the large sample size and range of analyses planned, effect size may be more meaningful than significance value.

We have added a discussion about this to the "Data Handling" section:

Page 7, Data Handling:

"Significance threshold:

As a general rule, we will set the alpha value at 0.05, as is common in the field. However, in some cases, a more conservative alpha value may be appropriate to reduce the risk of false positive results, for example, to account for multiple comparisons. We will also consider the potential impact of using different alpha values on the interpretation of our results. We will discuss our chosen alpha value in the methods section of any results papers. We also note that, given the large sample and range of analyses planned, effect size is likely to be more meaningful than significance value."

For analysis 4 in particular, will there be any consideration for accounting for multiple comparisons? The description of the mediation analyses can be expanded. Directionality and temporality are critical in mediation analysis. The authors should consider a directed acyclic graph to determine the position of the variables of interest in their model based on a priori and theoretical considerations.

Thank you to the reviewer for these questions and valuable suggestion to clarify this part of the manuscript.

We agree with the reviewer that in Analysis 4, given that we will be testing the associations between several measures of neurodegeneration and multiple cytokines, it is important to adjust for multiple comparisons to reduce the risk of false positive findings.

We plan to address the issue of multiple comparisons in several ways. First, we will use linear mixed models (LMM) to assess the relationship between measures of neurodegeneration and systemic inflammation while accounting for injury severity markers. LMM allows us to model repeated measures over time, and to account for within-subject correlation, thus reducing the number of statistical tests needed. Second, we will assess the predictive value of both cytokine clusters as well as individual 'discriminatory' cytokines. We will use a Bonferroni correction or another appropriate method to adjust for multiple comparisons in this analysis. Third, we will use mediation analysis to explore the extent to which relationships between injury characteristics and neurodegeneration are mediated by specific cytokines or cytokine clusters.

We agree that mediation analysis requires careful consideration of directionality and temporality of variables. A directed acyclic graph (DAG) can help us identify the different potential causal pathways between variables of interest and guide our mediation analysis. We will construct several DAGs to represent different theoretical models that match different temporal timepoints, with the variables of interest positioned in the appropriate order based on prior knowledge and theoretical considerations. The DAG will allow us to identify the potential direct and indirect effects of the variables of interest on the outcome, and to test the significance of the indirect effects using appropriate statistical methods.

In addition, we will conduct sensitivity analyses to assess the robustness of our findings to potential violations of the DAG assumptions, such as unmeasured confounding. We will report our DAG and sensitivity analysis results alongside main findings in any results papers.

We have expanded our description of the mediation analysis. We also include, as suggested, a DAG that illustrates our key hypotheses: 1) that injury severity and extent directly influence neurodegeneration, and 2) cytokine levels in the chronic period separately mediate the relationship between injury severity and extent and neurodegeneration. In this case, the mediation analysis will focus on the extent to which specific cytokines or cytokine clusters mediate the relationship between injury severity and extent and neurodegeneration. Depending on results from Analysis 1-3, other DAGs will also be constructed and assessed, for example, including cytokine levels at different timepoints.

Page 8, Data Analysis, Analysis 4:

"LMM enables modelling of repeated measures over time, and to account for within-subject correlation. This reduces the number of statistical tests needed, which reduces the need to correct for multiple comparisons. We will also use Bonferroni correction or another appropriate method to adjust for multiple comparisons in this analysis."

Page 8-9, Data Analysis, Analysis 4:

"We will construct directed acyclic graphs (DAG) to help us identify the different potential causal pathways between variables of interest and guide our mediation analysis. We will construct several

DAGs to represent different theoretical models that match different temporal timepoints, with the variables of interest positioned in the appropriate order based on prior knowledge and theoretical considerations. The DAG will allow us to identify the potential direct and indirect effects of the variables of interest on the outcome, and to test the significance of the indirect effects using appropriate statistical methods. Figure 3 shows a DAG illustrating our key hypotheses: 1) that injury severity and extent directly influence neurodegeneration, and 2) cytokine levels in the chronic period separately mediate the relationship between injury severity and extent and neurodegeneration. In this case, the mediation analysis will focus on the extent to which specific cytokines or cytokine clusters in the chronic period mediate the relationship between injury severity and extent and neurodegeneration (**FIGURE 3**). Further DAGs will be constructed based on findings from above analyses. In addition, we will conduct sensitivity analyses to assess the robustness of our findings to potential violations of the DAG assumptions, such as unmeasured confounding. We will report our DAG and sensitivity analysis results alongside main findings in any results papers.”

FIGURE 3:

FIGURE 3 legend:

In this directed acyclic graph, injury severity and extent are hypothesised to cause both early (up to 6 weeks post-TBI) and late (6+ months post-TBI) systemic inflammation, as well as directly influence neurodegeneration. Additionally, late systemic inflammation is hypothesised to partly mediate the relationship between injury severity and extent and neurodegeneration. Red denotes processes, blue indicates events and green denotes outcome measures. CT = computer tomography scan, WM = white matter, GM = grey matter, FA = fractional anisotropy.

Comments to the Author:

This is an important and well described study that leverages existing data and samples generated from two large prospective TBI studies. The protocol is thorough and appropriate to investigate the primary aims of the study. There are no apparent flaws that will prevent sound interpretation of the data.

One minor suggestion is for the introduction. This section provides the rationale for how systemic inflammation might be a mediator of neurodegeneration. How TBI can lead to systemic inflammation is not raised. The authors might consider outlining what is known or hypothesised about the primary cellular sources of raised blood cytokines after TBI.

Congratulations on nice protocol and exciting study. I look forward to seeing the results.

Thank you to Dr McDonald for the positive comments. We are pleased to include a bit more detail in the "Introduction" on the background of what is already known or suggested to be the primary cellular sources of blood cytokines after TBI.

Page 2, Introduction:

"Neurotrauma may cause direct peripheral immune infiltration through blood-brain barrier disruption⁹, and raised blood proinflammatory cytokines have been demonstrated after moderate-severe TBI¹⁰. Blood brain barrier breakdown after trauma may cause centrally derived cytokines to leak into the wider circulation, but changes in serum cytokines after TBI are not entirely attributable to this. There is evidence, largely from animal studies, that cytokine production from peripheral immune organs, particularly the spleen and thymus, contributes to a systemic inflammatory response¹¹. Change in sympathetic tone on lymphoid organs after TBI may also be a factor resulting in changes in peripheral cytokine profile¹²."

VERSION 2 – REVIEW

REVIEWER	Mader, Marius Stanford University School of Medicine
REVIEW RETURNED	11-Apr-2023
GENERAL COMMENTS	The authors have responded to my comments sufficiently. Good luck for the study!
REVIEWER	Meier, Timothy Medical College of Wisconsin, Neurosurgery
REVIEW RETURNED	02-May-2023
GENERAL COMMENTS	The authors have very thoroughly addressed all of my previous comments and I have no additional comments. I look forward to following the published results from this interesting and important study.